# Affection of Respiratory Muscles in ALS and SMA

**DOI:** 10.3390/jcm11051163

**Published:** 2022-02-22

**Authors:** Wiebke Hermann, Simona Langner, Maren Freigang, Stefanie Fischer, Alexander Storch, René Günther, Andreas Hermann

**Affiliations:** 1Department of Neurology, University Medical Center Rostock, University of Rostock, 18147 Rostock, Germany; wiebke.hermann@med.uni-rostock.de (W.H.); alexander.storch@med.uni-rostock.de (A.S.); 2Deutsches Zentrum für Neurodegenerative Erkrankungen (DZNE) Rostock/Greifswald, 18147 Rostock, Germany; 3Division of Pulmonology, Medical Dept I, University Hospital Carl Gustav Carus, Technische Universität Dresden, 01307 Dresden, Germany; simona.langner@uniklinikum-dresden.de; 4Department of Neurology, University Hospital Carl Gustav Carus, Technische Universität Dresden, 01307 Dresden, Germany; maren.freigang@uniklinikum-dresden.de (M.F.); stefanie.fischer@uniklinikum-dresden.de (S.F.); rene.guenther@uniklinikum-dresden.de (R.G.); 5Center for Transdisciplinary Neurosciences Rostock (CTNR), University Medical Center Rostock, University of Rostock, 18147 Rostock, Germany; 6Deutsches Zentrum für Neurodegenerative Erkrankungen (DZNE), 01307 Dresden, Germany; 7Translational Neurodegeneration Section “Albrecht-Kossel”, Department of Neurology, University Medical Center Rostock, University of Rostock, 18147 Rostock, Germany

**Keywords:** amyotrophic lateral sclerosis, spinal muscular atrophy, respiratory failure, diaphragm ultrasound, hypercapnia, hypoxemia

## Abstract

Respiratory dysfunction is a common cause of morbidity and mortality in motor neuron disease (MND). However, classical volitional measures of respiratory function in these patients are impeded by, e.g., bulbar paralysis or progressive disability. Diaphragm ultrasound imaging might be a valuable tool for assessing respiratory impairment, albeit different ultrasound measures have not been systematically investigated in adult MND patients and, in particular, have not yet been comparatively applied in adult patients with amyotrophic lateral sclerosis (ALS) and spinal muscular atrophy (SMA). We hypothesized that in contrast to ALS patients, adult SMA patients show a relative sparing of diaphragm function. We retrospectively analyzed diaphragm ultrasound imaging data of 40 patients with ALS and 23 patients with SMA in comparison to a multitude of established parameters of respiratory function. Indeed, ALS patients showed more severe diaphragm dysfunction than adult SMA patients, however, diaphragm dysfunction was also common in adult SMA patients. Notably, dynamic measures of diaphragm function rather than thickness measures were impaired in ALS compared to SMA. Thus, diaphragm ultrasound imaging might be a useful tool to evaluate respiratory dysfunction in adult MND patients. Future larger and prospective studies are needed to validate our initial findings.

## 1. Introduction

Respiratory dysfunction is a common cause of morbidity and mortality in many neuromuscular disorders, but particularly important in motor neuron diseases (MND), such as amyotrophic lateral sclerosis (ALS) or spinal muscular atrophy (SMA). While ALS comprises sporadic and genetic forms with its main prevalence in older ages, SMA is a genetic disease mainly appearing in early child- and young adulthood. Despite the fact that the genetics and molecular pathophysiology of ALS and SMA might be significantly different, they share a common clinical and neuropathological picture of motor neuron demise. Respiratory dysfunction is mainly due to lower motor neuron loss or dysfunction in both ALS and SMA (for review see [1]). In ALS, respiratory insufficiency inevitably occurs sooner or later over the disease course resulting in its high mortality. Non-invasive ventilation (NIV) has been proven to prolong survival in ALS patients. While this was particularly true for ALS patients with spinal onset, NIV increases the quality of life in both bulbar and spinal onset ALS [2,3,4]. Due to novel therapeutic options for spinal muscular atrophy (SMA), adult SMA patients are consulting adult MND specialist care more frequently and, therefore, ALS specialists are increasingly involved in the treatment of these patients. This might lead to different approaches towards respiratory dysfunction measurements and treatment in these patients, which has not been extensively studied, yet. Furthermore, treatment with gene-modifying therapy might result in the emergence of new phenotypes with still uncertain respiratory involvement [5,6].

In MND, respiratory failure is not caused by dysfunction of gas exchange such as in primary lung diseases, but mainly by impaired ventilation resulting from a complex interaction of inspiratory muscle weakness, inspiratory muscle fatigue, and constitutional abnormalities such as kyphoscoliosis [7,8]. The assessment of respiratory function is still up for debate and includes a variety of potential measurements, each of them with certain advantages and pitfalls [9,10]. Respiratory function is typically investigated by volitional lung function tests, such as vital capacity (VC), maximal inspiratory/expiratory pressure (MIP/MEP), and sniff nasal inspiratory pressure (SNIP). While many of these measures are well established and helpful in identifying respiratory failure, they all require full patient cooperation and therefore are of limited use in patients with progressed neuromuscular diseases. Common obstacles for VC measurement include insufficient mouth closure in the case of significant bulbar symptoms, whereas advanced disease stages with accumulated disability interfere with spirometry of body plethysmography, which is true for both ALS and SMA patients [11,12].

On the other hand, the involvement of thoracic muscles in MND—specifically in the rare cases of thoracic symptom onset—is difficult to assess. P 0.1 measurement in body plethysmography or comparison of vital capacity in sitting vs. supine position may indicate (dys-)function of respiratory muscles or the diaphragm itself, respectively. However, these are only indirect measurements of respiratory muscles function, further hampered by the above-mentioned pitfalls. Other techniques include diaphragm neurography, which is suitable for detecting reduced compound muscle action potential amplitude, and diaphragm myography proving denervation of these muscles [13]. However, these techniques are technically demanding, and myography of the diaphragm carries a significant risk of adverse events, such as inducing a pneumothorax. Furthermore, results of diaphragm myography are not standardized and quantifiable, thus they are not helpful in determining the decline of respiratory function. Diaphragm ultrasound has become increasingly recognized as a valuable tool to investigate both diaphragm structure and function [13]. It has the advantage of being harmless and easy to use. However, there have been surprisingly few studies of its value in the diagnosis of respiratory dysfunction in ALS, and none in adult SMA patients (for review see [14]). In addition, there is a lack of systematic comparison of the different diaphragm ultrasound imaging parameters with each other and with other established methods as well as signs of respiratory failure in MND [14].

We thus systematically analyzed both thickness/thickening fraction of the diaphragm as well as diaphragm excursion in adult MND patients and systematically compared them to a multitude of established measures of respiratory impairment. We hypothesized that both juvenile and adult SMA patients show relatively spared diaphragm function compared to ALS. Finally, we addressed the feasibility of diaphragm ultrasonography in adult ALS and SMA patients including advanced stages.

## 2. Materials and Methods

### 2.1. Patients and Study Design

In this monocentric study, we retrospectively analyzed data from adult MND patients who underwent routine clinical diagnostics concerning respiratory impairment due to MND as well as diaphragm ultrasound imaging at the Department of Neurology of the University Hospital Dresden from June 2017 to March 2021. These included individuals with definite, probable, or possible ALS according to the revised El Escorial criteria, as well as genetically proven 5q-associated spinal muscular atrophy (SMA). The study was part of a prospective registry approved by the institutional review boards (EK 49022016; EK 393122012), patients gave written consent prior to inclusion into this registry. Only patients with evaluable diaphragm ultrasound imaging were included in this study (*n* = 63). Clinical and respiratory data of these patients were retrieved from the registry, with some of the patients lacking isolated parameters such as respiratory measures. Participants were excluded if they were diagnosed with any other respiratory dysfunction/disease prior to MND onset or did not tolerate ultrasound examination without invasive or non-invasive ventilation.

### 2.2. Spirometry and Clinical Routine Data Evaluation

Spirometry was performed in a sitting position using MicroLab ML3500 Spirometer. Forced vital capacity (FVC), forced expiratory volume within the first second of expiration (FEV1), and peak expiratory flow (PEF) were recorded. Spirometric values were depicted according to Criée et al. [15] as a percentage of predictive value within the analysis instead of absolute values since these are well known to be influenced by age, sex, and height. Additionally, patients underwent routine daytime capillary blood gas analysis, with results depicted in SI units if not mentioned otherwise.

Additionally, established motor scores (Hammersmith Functional Motor Scale Expanded [HFMSE] [16] and Revised Upper Limb Module [RULM] [17]), and the revised ALS Functional Rating Scale (ALSFRS-R) [18] were assessed. ALSFRSR-10 represents the item dyspnoea, ALSFRSR-11 is the item orthopnoea/respiratory dysfunction related to sleep, and ALSFRSR-12 is the item mechanical ventilation. The different motor scores comprise several items rating different motor skills with higher scores indicating better function, respectively.

### 2.3. Ultrasound Examination of the Diaphragm

To systematically compare the different values obtained by diagnostic diaphragm ultrasound, we analyzed both sides separately using B mode and M mode diaphragm ultrasound including the following direct measures: thickness (B mode) and excursion (M Mode), and additionally calculated the thickening ratio (end-inspiratory thickness/end-expiratory thickness) and thickening fraction (end-inspiratory thickness − end-expiratory thickness)/end-expiratory thickness × 100%) (both B mode). The ultrasound examinations were performed with the patient lying in a supine position during breathing at rest (without NIV or IV ventilation) using the ultrasound scanner “Xario 200; TUS-X200” from Toshiba. The measurements were performed by a respiratory physician experienced in thoracic sonography blinded for the underlying diagnosis (SL). The following parameters were determined:Side-separated diaphragm thickness.

Diaphragm thickness was measured separately for the left and right side in B mode with a 7.5–10 MHz linear transducer placed over the apposition zone of the diaphragm on the thorax in the 8th–9th intercostal space between the anterior axillary and mid-axillary line. In the apposition zone, the diaphragm is represented as a three-layered structure: an echo-depleted central layer bounded by two echo-rich layers, the pleural and peritoneal diaphragmatic lines. Diaphragm thickness was measured from the pleural line to the peritoneal line (center-to-center or top edge-to-top edge or bottom edge-to-bottom edge) [19,20,21]. In the end-inspiratory position, values < 2 mm in the apposition zone indicate atrophy of the diaphragm [20,22,23]. Figure 1A shows the sonographic determination of diaphragm thickness (healthy subject).

2.Thickening ratio and thickening fraction of the diaphragm.

Diaphragm thickening ratio was determined as the quotient of end-inspiratory thickness and end-expiratory thickness. The diaphragm thickening fraction during quiet breathing was calculated as the percentage change in relative diaphragmatic thickness. A thickening fraction of < 20% was defined as abnormal [19,22].
$$thickening\;ratio=\frac{end-inspiratory\;thickness}{end-expiratory\;thickness}\;$$
$$thickening\;fraction\;\left[\%\right]=\frac{\left(end-inspiratory\;thickness-end-expiratory\;thickness\right)}{end-expiratory\;thickness}\times 100\%$$

3.Side-separated excursion of the diaphragm.

Assessment of diaphragm excursion was performed in a side-separated fashion using a 3.5 MHz convex transducer. The probe was placed in the last intercostal space between the anterior axillary and mid-axillary line or below the xiphoid with an angle of insonation of ≥70°. First, B mode was used to visualize the hemidiaphragm in the hepatic or splenic window. Then, using M mode, the amplitude of the craniocaudal excursion was recorded. This method allows real-time observation and quantification of excursion of the diaphragm; values < 10 mm indicate diaphragmatic dysfunction [21,22,24,25]. Figure 1B shows the sonographic measurement of diaphragm excursion (healthy subject).

### 2.4. Statistics

Statistics were performed using either SPSS Software (version 25; IBM, Inc., Chicago, IL, USA) or GraphPad Prism (Version 8.4.3; LLC). Group comparisons for categorical data were conducted using χ^2^ or Fisher’s exact test as appropriate. Group comparisons for ordinal and metric data were performed using student’s *t*-test or Mann-Whitney U test as appropriate. Pearson’s correlation test and multivariate linear or binary logistic regression modeling were performed to test the association of demographic and clinical candidate factors such as disease entity, body mass index (BMI), and age with sonographic measures. Pearson’s correlation coefficient |r| < 0.3 was considered a weak, |r| = 0.3–0.59 a moderate, and |r| ≥ 0.6 a strong agreement/correlation. All *P*-values were two-sided and values of less than 0.05 were deemed statistically significant (not adjusted for multiple comparisons due to the pilot character of the study).

## 3. Results

### 3.1. Sociodemographic and Clinical Data

We included 40 patients with ALS and 23 patients with 5q-associated SMA. The study participants were assessed for demographic and clinical characteristics (Table 1) showing a total ALSFRSR median of 31 points for ALS and 32 for SMA, respectively. SMA patients were—as expected—significantly younger and showed lower BMI, while there were no differences in sex and total ALSFRSR between groups. Interestingly, ALS patients scored significantly lower on the respiratory part of the ALSFRSR (questions 10–12).

### 3.2. Diaphragm Ultrasound Is Feasible Even in Advanced MND

Diaphragm ultrasound was successfully performed in 63 patients, including patients in advanced disease stages (ASLFRSR range 9–42; HFMSE 0–41 and RULM 0–37).

### 3.3. Diaphragm Dysfunction Is More Prevalent in ALS Than SMA

When comparing the numerical values of the ultrasound parameters between the ALS and the SMA subcohort, diaphragm thickening ratio, thickening fraction, as well as excursion, were significantly decreased in the ALS compared to the SMA subcohort, while no significant differences were detected regarding diaphragm thickness (Mann-Whitney U-test; Table 2, Figure 2A–C). We did not detect any association of diaphragm imaging parameters with sex (*p* ≥ 0.05; Mann-Whitney U-test) or age, but BMI was associated with diaphragm thickness (data not shown in detail). Therefore, adjustment to the candidate covariates sex, age, and BMI using multivariate linear regression analyses led to non-significant differences between both groups except for diaphragm excursion (Figure 2 & Table 2).

We then analyzed the rate of pathological findings—determined by predefined cut-offs used for clinical routine [19,20,22,23,25]—comparing the two disease subcohorts. While diaphragm thickness was abnormal in only a few ALS and SMA patients, thickening ratios were below the reference range in nearly all patients (Appendix A). Of note, while the latter did not differ between the ALS and the SMA subcohort, thickening fraction and diaphragm excursion differed significantly between the two subcohorts with higher rates of pathological findings in the ALS subcohort (Fisher’s exact test; Appendix A). Even more, thickening fraction and diaphragm excursion showed pathologically abnormalities in a very similar pattern (Fisher’s exact test; Appendix A). However, adjustment to the candidate covariates sex, age, and BMI using binary logistic regression showed no significant differences between both groups. This confirms that diaphragm dysfunction is present in adult SMA patients also.

### 3.4. Respiratory Dysfunction Occurs More Frequently in ALS

We further analyzed the differences in established respiratory measures comparing the two disease subcohorts, again adjusted for age, sex, and BMI. Although the total ALSFRSR score was similar, the respiratory subscore of the ALSFRSR was significantly lower in the ALS compared to the SMA subcohort (Figure 2D). This was also true for ALSFRSR item 10 (=dyspnea), but not for item 11 (=orthopnoe/respiratory dysfunction related to sleep) or item 12 (mechanical ventilation; Appendix A). In a sitting position, VC (*n* = 55; *p* = 0.035) and FEV1 (*n* = 39; *p* = 0.039) differed significantly between the subcohorts. In addition, daytime pCO2 was significantly higher in the ALS subcohort, while all other blood gas values were not if adjusted for age, sex, and BMI (Figure 2E).

### 3.5. Diaphragm Excursion Correlated Best with Classical Respiratory Measures

Based on these results demonstrating (i) feasibility of diaphragm ultrasound in both ALS and SMA patients and (ii) prevalent diaphragm abnormalities also in SMA though still to a slightly lesser extent than in the ALS subcohort, we aimed at generating preliminary hypotheses in regard to the potential value of diaphragm ultrasound in the clinical care of these patients. Therefore, we further analyzed the associations of diaphragm ultrasound imaging in comparison to established respiratory measures such as vital capacity, spirometry, respiratory subscore of the ALSFRSR, and daytime blood gases separately for ALS (Table 3, Appendix A) and SMA (Table 4, Appendix A) using partial correlations adjusted for age and BMI.

In ALS, diaphragm excursion correlated best to ALSFRSR respiratory items as well as VC and FEV1. While this was also seen to a lesser extent in SMA; thickness parameters also showed correlations in SMA which were not demonstrated in ALS. Larger prospective studies are warranted to further investigate these preliminary findings.

## 4. Discussion

Respiratory dysfunction is a common cause of morbidity and mortality in MND. Classical volitional measures of respiratory function are impeded by, e.g., bulbar paralysis or progressive disability of the patients. We showed here that diaphragm ultrasound imaging might be a useful tool for assessing respiratory function in MND patients in general, but also specifically in severely affected patients. Furthermore, most of the adult SMA patients and nearly all of the ALS patients showed diaphragm ultrasound imaging abnormalities.

Diaphragm ultrasound imaging has been described in several studies with good intra- and interrater reliability in B mode ultrasound, both in terms of thickness and thickening [14]. Interestingly, however, combined assessment of B mode and M mode ultrasound in MND patients was not often reported so far, lacking their direct diagnostic comparison [14]. In contrast to many of the previous studies, we systematically investigated both thickness and thickening of the diaphragm as well as diaphragm excursion in MND patients (*n* = 63). By doing so, we showed that diaphragm excursion during breathing at rest (tidal breathing) was the clinically most meaningful parameter in our sample, especially in the ALS subcohort. Only some ALS and some SMA patients formally showed diaphragm atrophy (defined by ≤2 mm end-inspiratory thickness), but in nearly all patients reduced contractibility (defined by reduced thickening ratio ≤ 1.5) was detected. While a thickening ratio cut-off of 1.39 was reported to be able to identify hypercapnic patients with high specificity and sensitivity [26], we cannot confirm these findings in our study cohort. In contrast, reduced diaphragm excursion (defined as ≤10 mm in normal ventilation) correlated closely with clinical and respiratory measures, fitting with Carrie et al. who reported a high sensitivity of decreased diaphragm excursion to predict impaired pulmonary function [27].

The second aim of our study was to systematically compare the two most common MND forms in adult specialized care, namely ALS and SMA, the latter being of recent importance since these patients enlist adult specialist care after the approval of the novel gene modulating therapies [5]. While a significant amount of smaller studies on diaphragm ultrasound in ALS have already been published (for review see [14]), to our knowledge only very few studies are available for SMA type 1 [28] and none for adult SMA. Interestingly, diaphragm dysfunction seems to be less severe in SMA but is still present in many adult SMA patients (Figure 2, Table 2, Appendix A). This is in contrast to prior studies investigating juvenile SMA patients showing prominent involvement of expiratory and intercostal muscles rather than the diaphragm [7,29]. Furthermore, in “sitters” and “walkers”, scoliosis can additionally and significantly impair pulmonary function [30]. The latter often hampers classical volitional measures, thus diaphragm ultrasound imaging might be still helpful in these cases.

Limitations of this study include mainly its retrospective design and the monocentric approach. We included only MND patients who underwent diaphragm ultrasound imaging in our analysis, and of those, we retrospectively retrieved all other respiratory and clinical measures available. This might overemphasize the feasibility of diaphragm ultrasound imaging while underestimating the feasibility of the other measures. Additionally, even though reporting on so far the largest cohort, the sample size is still small. We described the two subcohorts of ALS and SMA using descriptive statistics, while also trying to develop the first conclusions from our sample to generate hypotheses using inferential statistics. This clearly needs to be validated in larger prospective cohorts with a more pre-specified hypothesis, which our study might help to develop. Furthermore, we did not evaluate interrater reliability systematically. Therefore, further research on diaphragm ultrasound in multicenter and prospective trials should be encouraged to confirm our findings.

In summary, we provide evidence for the reasonable applicability of diaphragm ultrasound imaging in clinical use for the determination/assessment of respiratory dysfunction of MND patients. Dynamic measures, especially diaphragm excursion, showed closer associations with respiratory measurements than diaphragm thickness, which was specifically true in ALS. All measurements took place during breathing at rest. This further reduced the need for the patient to collaborate, enabling it also in severely affected patients.

## Figures and Tables

**Figure 1 jcm-11-01163-f001:**
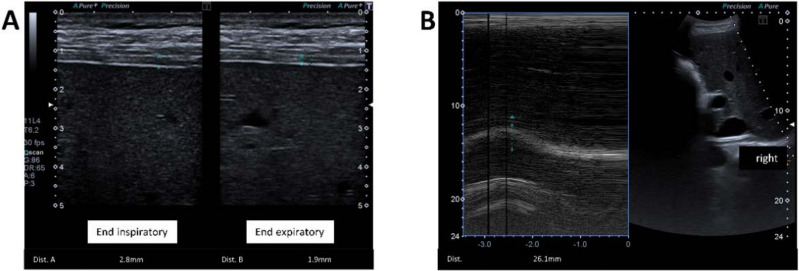
(**A**) Sonographically determined diaphragm thickness. Two-dimensional real-time mode with the end-inspiratory/-expiratory thickness of the right diaphragm. Sonographic window: midaxillary line at the level of the diaphragmatic apposition zone. Measurement points: diaphragmatic and peritoneal pleural lines. (**B**) Sonographically determined diaphragmatic excursion. M mode during quiet breathing in the supine position.

**Figure 2 jcm-11-01163-f002:**
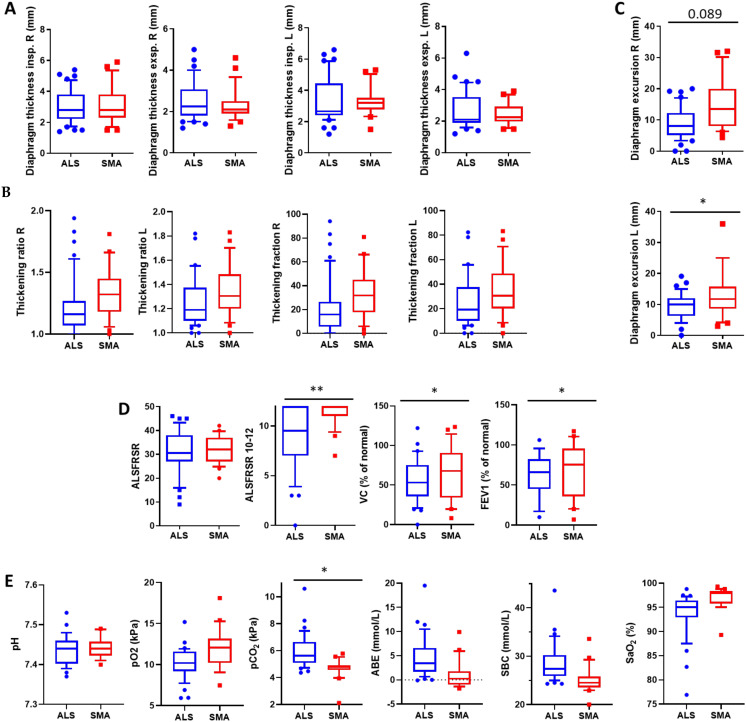
(**A**,**B**) Two-dimensional real-time mode with the end-inspiratory/-expiratory thickness of the diaphragm. Sonographic window: mid axillary line at the level of the diaphragmatic apposition zone. Measurement points: diaphragmatic and peritoneal pleural lines. Depicted are diaphragm thickness (**A**) and thickness ratio and fraction (**B**) in B mode, respectively. (**C**) Diaphragm excursion in M mode during breathing in a supine position at rest. (**D**) shows clinical and classical volatile respiratory measures, (**E**) daytime blood gas analysis results. Abbreviations: ALSFRSR: ALS functional rating scale revised; VC (%), vital capacity (in percent of normal value); FEV1 (%), the first second of expiration (in percent of normal value). Normative values of spirometry were calculated according to Criée and colleagues [15], ABE, actual base excess; SBC, standard base concentration. Shown are box blots with IQR, the whiskers set at 10–90% percentiles. Multivariate linear regression (for continuous outcome variables) analyses were performed to adjust for the candidate covariates sex, age, and BMI. * indicates *p* values < 0.05; ** *p* values < 0.01.

**Table 1 jcm-11-01163-t001:** Demographic and clinical characteristics of study cohorts.

Parameter/Score	Outcome	ALS (*n* = 40)	SMA (*n* = 23)	*p*-Value
**Age [year],** **median (IQR)**		66 (59–73)	32 (22–46)	<0.001 ^†^
**Sex,** n (%)	female male	18 (45%) 22 (55%)	14 (61%) 9 (39%)	0.297 ^‡‡^
**ALS type,** n (%)	bulbar spinal unknown	16 (40%) 23 (57%) 1 (3%)		
**SMA type,** n (%)	2 3		10 (44%) 13 (56%)	
***SMN2*****copy number,** n (%)	2 3 4 unknown		1 (4%) 13 (57%) 8 (35%) 1 (4%)	
**BMI [kg/m^2^],** **median (IQR)**		24 (23–28)	21 (19–26)	0.023 ^§^
**ALSFRSR [score],** **median (IQR)**		31 (27–38)	32 (27–37)	0.733 ^†^
**ALSFRSR respiratory function [ALSFRSR questions 10–12],** **median (IQR)**		10 (7–12)	12 (11–12)	<0.001 ^§^
**HFMSE [score],** **median (IQR)**			8 (4–32)	
**RULM [score],** **median (IQR)**			19 (14–35)	

Data are median (IQR—inter quartile range) or numbers (%) as appropriate. *p* Values are from ^§^ Mann-Whitney-U test, ^†^ student’s *t*-test-U-test, ^‡‡^ χ^2^ test or Fisher’s exact test. Abbreviations: ALS, Amyotrophic lateral sclerosis; SMA, spinal muscular atrophy; SMN, survival of motor neuron; BMI, body mass index; ALSFRSR, ALS functional rate scale revised; HFMSE, Hammersmith functional motor scale expanded; RULM, revised upper limb module.

**Table 2 jcm-11-01163-t002:** Differences of diaphragm ultrasound imaging between the ALS and SMA subcohorts.

Parameter	Outcome	ALS (*n* = 40)	SMA (*n* = 23)	*p*-Value ^§^	Multiple Linear Regression Analysis ^#^	
					Adjusted Correlation Coefficient (95% CI)	*p*-Value
**Diaphragm thickness insp. right**	median (IQR)	2.8 (2.2–3.8)	2.8 (2.3–3.8)	0.881	0.066 (–0.901–1.033)	0.892
**Diaphragm thickness insp. left**	median (IQR)	2.7 (2.4–4.5)	3.2 (2.8–3.5)	0.245	0.573 (–0.528–1.674)	0.301
**Diaphragm thickening ratio right**	median (IQR)	1.2 (1.1–1.3)	1.3 (1.2–1.5)	**0.006**	0.099 (–0.097–0.294)	0.317
**Diaphragm thickening ratio left**	median (IQR)	1.2 (1.1–1.4)	1.3 (1.2–1.5)	0.080	0.129 (–0.171–0.428)	0.393
**Diaphragm thickening fraction right**	median (IQR)	15.9 (5.5–26.5)	31.8 (17.6–45)	**0.006**	10.006 (–9.698–29.710)	0.313
**Diaphragm thickening fraction left**	median (IQR)	19 (10.0–37.5)	30.7 (20.2–48.8)	0.080	12.856 (–17.135–42.846)	0.393
**Diaphragm excursion right**	median (IQR)	8.1 (5.1–12.2)	13.5 (8.0–20.0)	**0.002**	5.138 (–0.778–11.054)	0.087
**Diaphragm excursion left**	median (IQR)	10.0 (6.3–12.0)	11.8 (8.6–15.8)	0.057	7.040 (1.458–12.621)	**0.015**

Depicted are median (IQR) values of diaphragm ultrasound parameters in both ALS and SMA. ^§^
*p* Values are from the Mann-Whitney-U test (numeric variables). ^#^ Multivariate linear regression (for continuous outcome variables) analyses were performed to adjust for the candidate covariates sex, age, and BMI. The reported correlation coefficient (B, 95% CI) > 0 indicates the magnitude of the positive correlation of the disease entity (ALS vs. SMA) on the respective diaphragm parameter after adjustment for sex, age, and BMI. Bold indicates *p* values < 0.05.

**Table 3 jcm-11-01163-t003:** Correlation of diaphragm B and M mode ultrasound parameters and classical measures of respiratory dysfunction in ALS patients.

	Thickness R Insp	Thickness R Exsp	Thickness L Insp	Thickness L Exsp	Thickening Ratio R	Thickening Ratio L	Thickening Fraction R	Thickening Fraction L	Excursion R	Excursion L	
**ALSFRSR total**	−0.17	−0.26	−0.29	−0.31	0.22	−0.05	0.23	−0.05	0.32	**0.48**	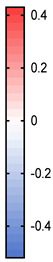
**ALSFRSR 10**	−0.11	−0.16	**−0.41**	**−0.38**	0.09	−0.10	0.10	−0.10	0.28	**0.42**	
**ALSFRSR 11**	−0.21	−0.27	−0.33	−0.31	0.16	−0.13	0.17	−0.13	0.22	0.24	
**ALSFRSR 12**	−0.01	−0.06	−0.07	−0.15	0.04	0.06	0.05	0.06	0.19	**0.43**	
**ALSFRSR 10–12**	−0.13	−0.19	−0.33	−0.33	0.12	−0.08	0.12	−0.08	0.27	**0.40**	
**FEV1 percent**	0.16	0.01	−0.20	−0.19	0.38	−0.31	0.35	−0.31	0.11	**0.57**	
**VC percent**	0.07	−0.14	0.16	0.10	**0.47**	0.02	**0.47**	0.02	0.25	**0.48**	
**pH**	−0.15	−0.21	−0.04	−0.12	0.13	0.24	0.14	0.24	0.19	0.18	
**pO_2_**	0.07	0.06	−0.07	−0.10	0.06	0.22	0.05	0.22	0.17	0.14	
**pCO_2_**	0.17	0.31	0.16	**0.40**	−0.27	**−0.40**	−0.28	**−0.40**	−0.33	−0.29	
**sO_2_**	−0.06	−0.13	−0.17	−0.22	0.16	0.19	0.16	0.19	0.32	0.27	
**ABE**	0.08	0.21	0.15	0.38	−0.25	−0.34	−0.25	−0.34	−0.28	−0.23	
**SBC**	0.12	0.25	0.18	**0.41**	−0.26	−0.34	−0.26	−0.34	−0.28	−0.23	

Depicted are Pearson’s correlation coefficient r for partial correlations adjusted for age and BMI; significant values are marked in bold (*p* < 0.05. two-sided). Abbreviations: ALSFRSR—ALS functional rating scale revised; VC (%)—vital capacity (in percent of normal value); FEV1 (%)—the first second of expiration (in percent of normal value). Normative values of spirometry were depicted according to Criée et al. [15]. ABE—actual base excess; SBC—standard base concentration. For more details see Appendix A.

**Table 4 jcm-11-01163-t004:** Correlation of diaphragm B and M mode ultrasound parameters and classical measures of respiratory dysfunction in SMA patients.

	Thickness R Insp	Thickness R Exsp	Thickness L Insp	Thickness L Exsp	Thickening Ratio R	Thickening Ratio L	Thickening Fraction R	Thickening Fraction L	Excursion R	Excursion L	
**ALSFRSR total**	−0.14	−0.13	−0.37	−0.52	−0.15	0.38	−0.15	0.37	** 0.44 **	0.19	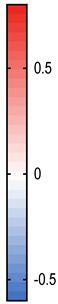
**ALSFRSR 10**	0.23	0.16	0.04	−0.03	0.29	0.13	0.29	0.13	0.26	0.07
**ALSFRSR 11**	−0.08	−0.03	−0.31	−0.33	−0.16	0.05	−0.16	0.05	** 0.49 **	0.12
**ALSFRSR 12**	−0.05	0.01	−0.28	−0.44	−0.21	0.25	−0.21	0.25	0.43	0.17
**ALSFRSR 10–12**	0.02	0.05	−0.31	−0.44	−0.09	0.22	−0.09	0.22	** 0.61 **	0.19
**HFSME**	−0.16	−0.21	−0.38	−0.45	−0.05	0.15	−0.06	0.15	0.12	0.07
**RULM**	−0.04	−0.06	−0.25	−0.24	−0.05	0.01	−0.05	0.01	0.17	0.06
**FEV1 percent**	−0.31	−0.31	** −0.53 **	** −0.56 **	−0.19	0.16	−0.20	0.16	0.38	0.30
**VC percent**	−0.23	−0.28	** −0.57 **	** −0.51 **	−0.08	0.00	−0.08	0.00	0.38	0.21
**pH**	** 0.61 **	** 0.80 **	** 0.56 **	** 0.55 **	−0.05	−0.11	−0.05	−0.11	0.02	−0.02
**pO_2_**	0.11	0.24	−0.01	−0.09	−0.24	0.10	−0.24	0.10	0.26	−0.05
**pCO_2_**	** −0.47 **	** −0.60 **	** −0.50 **	** −0.51 **	0.01	0.09	0.01	0.09	−0.15	−0.13
**sO_2_**	−0.13	−0.23	−0.27	−0.40	0.03	0.23	0.03	0.23	0.24	0.10
**ABE**	**0.59**	** 0.79 **	0.46	0.45	−0.07	−0.16	−0.07	−0.16	−0.20	−0.24
**SBC**	0.23	0.28	0.22	0.22	0.02	−0.07	0.02	−0.07	** −0.50 **	−0.15

Depicted are Pearson’s correlation coefficient r of partial correlations adjusted for age and BMI; significant values are marked in bold (*p* < 0.05. two-sided). Abbreviations: ALSFRSR—ALS functional rating scale revised; HFMSE—Hammersmith Functional Motor Scale Expanded; RULM—Revised Upper Limb Module; VC (%)—vital capacity (in percent of normal value); FEV1 (%)—the first second of expiration (in percent of normal value). Normative values of spirometry were depicted according to Criée et al. [15]; ABE—actual base excess; SBC—standard base concentration. For more details see Appendix A.

## Data Availability

All data is within the published version of the article.

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
