# Peer review of "Affection of Respiratory Muscles in ALS and SMA"

_jcm, 2022, doi:10.3390/jcm11051163_

Round 1

Reviewer 1 Report

In this study, Dr. Hermann and colleagues evaluated respiratory function determined by diaphgragm ultrasound imaging in 41 ALS patients and 23 SMA patients. They find respiratory dysfunction is more common in ALS than in SMA, and that ultrasound imaging can help diagnose respiratory dysfunction in adult MND patients. This is well-performed study with excellent data analysis. A major deficiency is the insufficient description and literature citation. Specific comments are below.

  1. In the introduction section, the authors might want to expand on pathological distinctions and similarities between SMA and ALS to provide readers with more background knowledge.
  2. The author stats this study is “a large cohort of MN patients” in line 83 and line 255, while the sample size is only 64 which might not be considered as “large” in cohort studies.
  3. Table 1 shows that SMA and ALS groups are significantly different in age. Could the diaphragm ultrasound imaging readout or the ALSFRSR respiratory subscore be affected by age?
  4. Line 44-46, the sentence “Due to novel therapeutic options for spinal muscular atrophy (SMA), the number of adult SMA patients is increasing and therefore ALS specialists are increasingly involved in treatment of these patients” is unclear. Why is it that the number of adult SMA patients and ALS specialists participating is increasing as a result of new treatment?
  5. Reference literatures are needed for below statements:
  • Line 49-50, “the gene-modifying therapy…”
  • Line 61-63 “Common obstacles…”
  • Line 69-73 “Other techniques include…”
  • Line 250-251, “Diaphragm ultrasound imaging has been described in several studies…”
  1. Typo in line 162, “was reported be able to” should be “was reported to be able to”

Author Response

In this study, Dr. Hermann and colleagues evaluated respiratory function determined by diaphgragm ultrasound imaging in 41 ALS patients and 23 SMA patients. They find respiratory dysfunction is more common in ALS than in SMA, and that ultrasound imaging can help diagnose respiratory dysfunction in adult MND patients. This is well-performed study with excellent data analysis. A major deficiency is the insufficient description and literature citation. Specific comments are below.

Response: We deeply thank the reviewer for this very positive and encouraging review.

  1. In the introduction section, the authors might want to expand on pathological distinctions and similarities between SMA and ALS to provide readers with more background knowledge.

Response: We added a more comprehensive comparative introduction for both disease entities.

  1. The author states this study is “a large cohort of MN patients” in line 83 and line 255, while the sample size is only 64 which might not be considered as “large” in cohort studies.

Response: We rephrased this properly. Nevertheless, it is the largest cohort so far, in which diaphragma ultrasonography in ALS/SMA patients is investigated.

  1. Table 1 shows that SMA and ALS groups are significantly different in age. Could the diaphragm ultrasound imaging readout or the ALSFRSR respiratory subscore be affected by age?

Response: We appreciate this comment. Age was different in between the groups, which lies in the nature of these two different presentations of a MND. Age at onset is different in SMA and ALS, thus we cannot match the patients except we would focus on atypical cases in one of the diseases. We still want to present the results of these two identities since these represent the two largest cohort in centers, who take care of adult MND patients nowadays.

Of note, we added statistical analyses including multiple linear regression and binary regression analysis to control for confounding covariates such as age, BMI and sex. Age did not correlate with any of the values except for diaphragma excursion on the right side. However, these effects did not persist in linear regression analysis, thus proofing that age is not a relevant cofactor in diaphragm ultrasound findings in our cohort.

  1. Line 44-46, the sentence “Due to novel therapeutic options for spinal muscular atrophy (SMA), the number of adult SMA patients is increasing and therefore ALS specialists are increasingly involved in treatment of these patients” is unclear. Why is it that the number of adult SMA patients and ALS specialists participating is increasing as a result of new treatment?

Response: We apologize, if we have been not clear enough here. The numbers of SMA patients are increasing (at least in the German adult MND centers) since the novel therapuetical options are available, due to the fact that before availability of these treatment options adult SMA patients were not treated by MND specialist, when reaching adulthood. We tried to improve the wording in the revised version of the manuscript.

  1. Reference literatures are needed for below statements:
  • Line 49-50, “the gene-modifying therapy…”
  • Line 61-63 “Common obstacles…”
  • Line 69-73 “Other techniques include…”
  • Line 250-251, “Diaphragm ultrasound imaging has been described in several studies…”

Response: We thank the reviewer for this comment and added respective literature

  1. Typo in line 262, “was reported be able to” should be “was reported to be able to”

Response: We corrected this typo.

Reviewer 2 Report

The manuscript “Differential affection of respiratory muscles in ALS and SMA” reported a method using diaphragm ultrasound imaging to help evaluate respiratory dysfunction of adult patients with amyotrophic lateral sclerosis (ALS) and spinal muscular atrophy (SMA). This retrospective study elicited the value of diaphragm ultrasound imaging as a novel clinical indicator. However, this work still has some aspects to be addressed.

Q1: Table 3 could be transformed into heat map, which could be clearer.

Q2: Did you exclude patients of underlying diseases with respiratory dysfunction?

Q3: The work did not consider the effect of age between ALS and SMA patients, whose age of onset is differential a lot. It is recommended to match the age of two diseases.

Author Response

The manuscript “Differential affection of respiratory muscles in ALS and SMA” reported a method using diaphragm ultrasound imaging to help evaluate respiratory dysfunction of adult patients with amyotrophic lateral sclerosis (ALS) and spinal muscular atrophy (SMA). This retrospective study elicited the value of diaphragm ultrasound imaging as a novel clinical indicator. However, this work still has some aspects to be addressed.

Response: We deeply thank the reviewer for his overall positive evaluation. As shown below, we carefully adressed every single open question.

Q1: Table 3 could be transformed into heat map, which could be clearer.

Response: We appreciate the reviewer’s suggestions a lot and now depict table 3 as a heat map. We additionally show the correlation coefficients and marked significant values in this heatmap providing as much information as possible for the interested reader on a now clearer and more memorabletable.

Q2: Did you exclude patients of underlying diseases with respiratory dysfunction?

Response: We appologize for not mentioning previously. We indeed did exclude patients suffereing from a previously established diagnosis of different respiratory dysfunction than MND. We now mention this clearly in the revised manuscript.

Q3: The work did not consider the effect of age between ALS and SMA patients, whose age of onset is differential a lot. It is recommended to match the age of two diseases.

Response: We appreciate this comment. Age might have an effect on respiratory functioning. However, age at onset is different in SMA and ALS, thus we cannot match the patients except we would focus on atypical cases in one of the diseases. We still want to present the results of these two identities since these represent the two largest cohorts in centers who take care of adult MND patients nowadays. We however tried to show the results more exploratively (see comments for reviewer 3). In addition, we clearly note this in the revised manuscript. Importantly, age did not correlate with any of the values except for diaphragma excursion on the right sight. Therefore we also included linear regression models which show that age is not a relevant covariate to explain our results.

Reviewer 3 Report

In this study, Hermann et al assess the role of several diaphragm ultrasound measures in patients with SMA and ALS. The main result is that the diaphragm excursion might be an interesting measure in these patients. While this is an interesting field, the study is not methodologically sound and the results are not adequately displayed, limiting its clinical interpretation, and meaning. Some major pitfalls:

  1. It is not stated if patients signed informed consent or not. If not, please, justify, since diaphragm ultrasound is, up to now, not part of the routine clinical practice.
  2. Several respiratory measures are reported. However, it is not stated if all patients performed all tests (including US), or how many patients performed each test), what could certainly affect the results of correlations. It is neither mentioned when were these measures acquired (the same day than US measures?). Moreover, they are not mentioned at all in the methodology.
  3. “For correlations Spearman-rank was used, multivariate linear regression modelling was performed to determine correlations of sonographic measures and candidate covariates. κ or Spearman’s rho correlation coefficient |r|<0.3 was considered a weak, κ/|r|=0.3–0.59 a moderate, κ/|r|≥0.6 a strong agreement/correlation”. Can really a correlation of 0.3 be considered as moderate? Moreover, linear regression allows to study associations, not correlations. Finally, although correlations can be used to generate hypothesis, I don’t think that performing a multiple linear regression of each variable is a good strategy or add some new information at all. Multiple regressions should be used to confirm associations (after adjusting by covariables) not to generate hypotheses.
  4. For merely descriptive purposes (e.g. Table 1, Table 2…), there is no need and it is not sound to use statistical tests. Moreover, the use of so many tests would require adjustment for multiple testing, so I suggest to just comparatively describe both populations (without tests). For example, in ALS population predominate male individuals and in SMA population, female ones, regardless of if this is statistically significant or not (the statistical analysis is irrelevant here). Furthermore, Table 2 is probably unnecessary, and it would be better just to display in the text the percentages of measures in the pathological range for each disease (including those of expiratory thickness, which are not shown in the table).
  5. Table 1 supposedly displays the median values and its IQR, but the IQR should be a range (i.e. two numbers), and not a single number as can be found in Table 1.
  6. The range of ALSFRS-R and ALSFRS-R respiratory function includes patient(s) with 0 points. This corresponds to patients who require continuous invasive ventilation. Any patient requiring ventilation during the US examination should have been excluded of the study, since US measurements would reflect the effect of ventilation rather that the actual ability of patients.
  7. It is for me impossible to understand how the diaphragm thickening ratio and fraction can have such a low (or even not at all) correlation with the thickness exp and thickness insp, if the former are calculated with the latter. This is just impossible unless all calculations are wrong.
  8. Table 3 is unintelligible. It should be substituted by a correlation matrix. Moreover, correlations for ALS patients and for SMA patients, should be given and interpreted separatedly.
  9. VC and FEV1 values are displayed in litres. Since baseline characteristics of SMA and ALS patients differ (age, height, wheight), in order to be comparable, these measures should be displayed as a % of normality.
  10. “VC correlated with none of the daytime blood gas analysis parameters” Other studies (e.g. Manera et al JNNP, etc…) have shown a correlation between FVC and BGA in ALS patients. Why this study fails to find that?
  11. The results of multiple linear regression analysis are only partially displayed, making them uninterpretable. Moreover, as I commented above, I would use them to confirm hypotheses or to respond some questions. If the intention is merely descriptive, they should be removed because they add no information to simple correlations.
  12. “the [use of] ALSFRSR should be considered [in SMA] based on our findings”. This is not at all supported by the data, since no correlation between ALSFRS-R and respiratory measures was found in the SMA population (this correlation was made with the whole cohort of patients.
  13. There are much more limitations than those acknowledged in the discussion.
  14. Overall, many statements in the abstract and discussion should be toned down. E.g. this study does not “provide evidence for good applicability of diaphragm ultrasound imaging in routine clinical use for the determination / assessment of respiratory dysfunction of MND patients”. Its incorporation into the clinical practice would require much more (and better) data than those provided here.

Minor:

  1. In methodology: remove the sentence “Demographic and clinical characteristics of the study populations are displayed in Table 1.” This belongs to the results section.
  2. In results:
    1. “SMA patients were – as expected – significantly younger and smaller”. I don’t think that smaller is the appropriate word here.
    2. “To systematically compare the different values obtained by diagnostic diaphragm 165 ultrasound, we analyzed both sides separately using B mode and M mode diaphragma 166 ultrasound including the following direct measures: thickness (B mode) and excursion (M 167 Mode) and additionally calculated thickening ratio (end-inspiratory thickness/end-expiratory thickness) and thickening fraction ((end-inspiratory thickness - end-expiratory thickness) / end-expiratory thickness x 100%) (both B mode).” This belongs to methodology.
    3. Please, use height instead of size.
  3. Overall, it would benefit of some English revision.

Author Response

In this study, Hermann et al assess the role of several diaphragm ultrasound measures in patients with SMA and ALS. The main result is that the diaphragm excursion might be an interesting measure in these patients. While this is an interesting field, the study is not methodologically sound and the results are not adequately displayed, limiting its clinical interpretation, and meaning. Some major pitfalls:

Response: We deeply thank the reviewer for the very constructive and helpful comments. As depicted below, we did adress every single concern raised. Of note, the other two reviewers were quite enthusiastic about our study. Thus we tried our best to address the questions of all three reviewers properly.

  1. It is not stated if patients signed informed consent or not. If not, please, justify, since diaphragm ultrasound is, up to now, not part of the routine clinical practice.

Response: As it was already written in the „Institutional Review Board Statement“ and „Informed Consent Statement“: The patients gave of course informed consent. We now additionally include such a sentence in the methods section pf the reivesed version of the manuscript (were the ethical aproval was already depicted though).

  1. Several respiratory measures are reported. However, it is not stated if all patients performed all tests (including US), or how many patients performed each test), what could certainly affect the results of correlations. It is neither mentioned when were these measures acquired (the same day than US measures?). Moreover, they are not mentioned at all in the methodology.

Response: We deeply thank the reviewer for raising this point. We included only patients who underwent diaphragma ultrasound imaging (n=100%), and of those we retrospectively retrieved all other respiratory and clinical measures available. We now mention this in more detail in the revised version of the manuscript (methodology, results, discussion) and do discuss potential pitfalls resulting from this. We additionally add proper descriptions of these measures in the methodology and like to apologize for not having done this already in the previous version.

  1. “For correlations Spearman-rank was used, multivariate linear regression modelling was performed to determine correlations of sonographic measures and candidate covariates. κ or Spearman’s rho correlation coefficient |r|<0.3 was considered a weak, κ/|r|=0.3–0.59 a moderate, κ/|r|≥0.6 a strong agreement/correlation”. Can really a correlation of 0.3 be considered as moderate?

Response: We cknowledge the reviewer’s opinion! Of course, the exact values of weak-moderate-strong correlations can be discussed. Therefore our intention was and still is to show all data, that the interested reader him- or herself can made his decision on the value of the respective correlation coefficient. Nevertheless, we used the above mentioned definition since it is widely used in the literature (e.g.:  Cicchetti DV. Multiple comparison methods: establishing guidelines for their valid application in neuropsychological research. J Clin Exp Neuropsychol 1994;16(1):155-161).

  1. Moreover, linear regression allows to study associations, not correlations. Finally, although correlations can be used to generate hypothesis, I don’t think that performing a multiple linear regression of each variable is a good strategy or add some new information at all. Multiple regressions should be used to confirm associations (after adjusting by covariables) not to generate hypotheses.

Response: We deeply thank the reviewer for his in depth review and discussion on our statistical methods (see also following points). While we agree with the reviewer that our study first of all is about a comparatively description of the both populations. Nevertheless, we were interested in understanding the value of the respective diaphragma ultrasound imaging values compares to classical respiratory measures. This is why we show the multiple regressions and still think that these add significant new value to the manuscript, which is also underpinned by the comments of reviewer 1&2.

  1. For merely descriptive purposes (e.g. Table 1, Table 2…), there is no need and it is not sound to use statistical tests. Moreover, the use of so many tests would require adjustment for multiple testing, so I suggest to just comparatively describe both populations (without tests). For example, in ALS population predominate male individuals and in SMA population, female ones, regardless of if this is statistically significant or not (the statistical analysis is irrelevant here).

Response: Since we are comparing the two cohorts, the corresponding statistical group comparison is indispensable

  1. Furthermore, Table 2 is probably unnecessary, and it would be better just to display in the text the percentages of measures in the pathological range for each disease (including those of expiratory thickness, which are not shown in the table).

Response: We do acknowledge the reviewer’s point, but we still feel that it is helpful for the daily work as neurologists with diaphragma ultrasound results. Thus we’d like to suggest to keep the modified version of the table in the manuscript.

  1. Table 1 supposedly displays the median values and its IQR, but the IQR should be a range (i.e. two numbers), and not a single number as can be found in Table 1.

Response: We did change it to the range

  1. The range of ALSFRS-R and ALSFRS-R respiratory function includes patient(s) with 0 points. This corresponds to patients who require continuous invasive ventilation. Any patient requiring ventilation during the US examination should have been excluded of the study, since US measurements would reflect the effect of ventilation rather than the actual ability of patients.

Response: We do appreciate this comment a lot. We carefully reanalyzed the complete sample. Indeed, there were only a few invasive ventilated patients. Of those, all but one were not continuously ventilated invasively (which however still leads to „0“ in item 12 of the ALSFRSR). Thus those, who were without ventilation during the US measurements were still included in the analysis, but one patient was removed since we could not for 100% making sure, that he might have been at the machine during measurement (even though this is unlikely since the examiner (=pulmologist) is of course well aware of this issue). Because of this, we consequently adopted all data throughout the whole manuscript.

  1. It is for me impossible to understand how the diaphragm thickening ratio and fraction can have such a low (or even not at all) correlation with the thickness exp and thickness insp, if the former are calculated with the latter. This is just impossible unless all calculations are wrong.

Response: We deeply thank the reviewer for his statistical advice. In the initial manuscript we presented spearman’s correlation, which represents rank scores rather than direct correlations. However, thickening ratio and fraction are already calculated compounds, which might influence correlation statistics, To improve the statistical soundness in regard to the reviewer’s comment, we addressed his/her concern by using the direct correlations of pearsons’s, leading to a clearer picture regarding these measures.

  1. Table 3 is unintelligible. It should be substituted by a correlation matrix. Moreover, correlations for ALS patients and for SMA patients, should be given and interpreted separatedly.

Response: As mentioned above, we wanted to depict the values for the interested reader that he/she can make his/her own conclusions about the value of the respective correlation coefficient. Thus we tried hard to improve readability. In addition, we additionally provide the separate tables for ALS and SMA in the new supplement

  1. VC and FEV1 values are displayed in litres. Since baseline characteristics of SMA and ALS patients differ (age, height, wheight), in order to be comparable, these measures should be displayed as a % of normality.

Response: We thanks the reviewer for this great suggestions. We did now use the % of normal value throughout the manuscript.

  1. “VC correlated with none of the daytime blood gas analysis parameters” Other studies (e.g. Manera et al JNNP, etc…) have shown a correlation between FVC and BGA in ALS patients. Why this study fails to find that?

Response: Using the % of normal values we now show nice and expected correlations of VC.

  1. The results of multiple linear regression analysis are only partially displayed, making them uninterpretable. Moreover, as I commented above, I would use them to confirm hypotheses or to respond some questions. If the intention is merely descriptive, they should be removed because they add no information to simple correlations.

Response: As mentioned above (comment 3 - 4), we (and the other reviewers) still believe that the multiple regressions are of interest for the reader and necessary to interprete our results. Thus we even elaborated on them that these are also easier to follow in the revised manuscript now easy to understand.

  1. “the [use of] ALSFRSR should be considered [in SMA] based on our findings”. This is not at all supported by the data, since no correlation between ALSFRS-R and respiratory measures was found in the SMA population (this correlation was made with the whole cohort of patients.

Response:  We disagree with the reivewer. We now show the correlations also seperated for the different subcohorts in the supplemental material. As shown tehre, the ALSFRSR respiratory subscore still correlated with diaphragma excursion. Even though we did see less correlations of the ALSFRSR respiratory subscore in the sub-cohort of SMA patients, it is remarkable that the highest correlation was shown exactly in this score. Thus, there is need to add a sensitive scoring system for respiratory dysfunction also in patients with SMA for routine monitoring, which might be the ALSFRSR respiratory subscore, Nevertheless, we toned this down in the revised version of the manuscript.

  1. There are much more limitations than those acknowledged in the discussion.

Response: We elaborated on this point.

  1. Overall, many statements in the abstract and discussion should be toned down. E.g. this study does not “provide evidence for good applicability of diaphragm ultrasound imaging in routine clinical use for the determination / assessment of respiratory dysfunction of MND patients”. Its incorporation into the clinical practice would require much more (and better) data than those provided here.

Response:  We appologize that we might have been misunderstood. Thus we toned down in the revised version of the manuscript. Nevertheless it is of note, that we present the so far largest cohort of SLA and SMA patients who were systematically investigated in B- and M-mode diaphragma ultrasound imageing and additionally correlated systematically to classical clinical and respiratory parameters. As reviewer 1&2 noted, we do believe that our study add relevant novel insights and additional evidence for the usability and validity of diaphragma ultrasound imaging in MND.

Minor:

  1. In methodology: remove the sentence “Demographic and clinical characteristics of the study populations are displayed in Table 1.” This belongs to the results section.

Response: we removed this in the methods section

  1. In results:
    1. “SMA patients were – as expected – significantly younger and smaller”. I don’t think that smaller is the appropriate word here.

Response: We thank the reviewer and corrected this mistake

    1. “To systematically compare the different values obtained by diagnostic diaphragm 165 ultrasound, we analyzed both sides separately using B mode and M mode diaphragma 166 ultrasound including the following direct measures: thickness (B mode) and excursion (M 167 Mode) and additionally calculated thickening ratio (end-inspiratory thickness/end-expiratory thickness) and thickening fraction ((end-inspiratory thickness - end-expiratory thickness) / end-expiratory thickness x 100%) (both B mode).” This belongs to methodology.

Response: we moved this to the methodology

    1. Please, use height instead of size. (muss ich noch machen)

Response: We did

  1. Overall, it would benefit of some English revision.

Response: We performed a thorough English revision.

Round 2

Reviewer 1 Report

The authors adequately addressed my questions. 

Author Response

Response: We thank the reviewer for his final approval

Reviewer 2 Report

My questions and suggestions were well answered.

Author Response

(The authors gave the same response as above.)

Reviewer 3 Report

I appreciate the detailed response of the reviewers. They have addressed some of my concerns and accordingly improved the manuscript. Nevertheless, I still have serious concerns regarding the data acquisition and analysis:

  • The need of NIV during the US examination should have been an exclusion criteria, since it invalidates the measurements. Since US was performed with the patient lying, any patient scoring 1 point or less in items 10, 11 and 12 of ALSFRS-R (i.e. 3 points or less in the respiratory domain), should have been excluded (because they need NIV during examination). Authors should clarify if any patient was using NIV during examination and how do they score ALSFRS-R, because there is no way that a patient needing continuous NIV can tolerate the supine position (even less an ALS patient).
  • Authors do not distinguish between descriptive and inferential statistics. This is a common mistake, but it is still wrong. The former allows you to raise hypotheses that can be confirmed (or not) with the latter. Most of this work is descriptive and there is no need for such a large amount of statistical analysis (see my first report). For example, when authors describe the demographic characteristics of SMA and ALS patients they are not confirming any pre-specified hypothesis. So there is no need for statistical analysis. Moreover, the use of so many tests through the manuscript (without pre-specified hypotheses) would require the use of some kind of adjustment.
  • Since this is a retrospective study and not all patients performed all measurements, it should be clearly stated how many patients performed each measurement and when (was this on the same day of the US?). Otherwise, some results (such as correlations) cannot be adequately interpreted.
  • The main conclusion "we provide evidence for good applicability of diaphragm ultrasound imaging in routine clinical use for the determination / assessment of respiratory dysfunction of MND patients" is not supported by the data. Authors provide evidence for the viability of US imaging, but not for its "routine clinical use for the assessment of respiratory dysfunction". This is a mainly descriptive retrospective work that allows making some interesting hypotheses that should be confirmed in future prospective studies. I would overall recommend the authors to tone down their enthusiasm and to deepen the methodological limitations of the study, which are not sufficiently described.
I am aware that the authors may not agree with some of the points exposed above and I sincerely appreciate their effort to improve their manuscript. However, I consider that there are still serious methodological flaws.

Author Response

Comments and Suggestions for Authors

  • I appreciate the detailed response of the reviewers. They have addressed some of my concerns and accordingly improved the manuscript. Nevertheless, I still have serious concerns regarding the data acquisition and analysis.

Response: We appreciate the positive view of reviewer 3 on our revisions (similar to reviewers 1 and 2, who are satisfied with our manuscript and rated its current form as sufficient for publication) and addressed the remaining concerns in the following sections.

  • The need of NIV during the US examination should have been an exclusion criteria, since it invalidates the measurements. Since US was performed with the patient lying, any patient scoring 1 point or less in items 10, 11 and 12 of ALSFRS-R (i.e. 3 points or less in the respiratory domain), should have been excluded (because they need NIV during examination). Authors should clarify if any patient was using NIV during examination and how do they score ALSFRS-R, because there is no way that a patient needing continuous NIV can tolerate the supine position (even less an ALS patient).

Response: As already mentioned in our first point-by-point reply, there was no patient on NIV or IV during US. Nevertheless if they used NIV, it is appropriate to score them 3 points or less in the respective ALSFRSR sub-items. We added a respective comment now also for the NIV ventilated patients in the revised manuscript.

  • Authors do not distinguish between descriptive and inferential statistics. This is a common mistake, but it is still wrong. The former allows you to raise hypotheses that can be confirmed (or not) with the latter. Most of this work is descriptive and there is no need for such a large amount of statistical analysis (see my first report). For example, when authors describe the demographic characteristics of SMA and ALS patients they are not confirming any pre-specified hypothesis. So there is no need for statistical analysis. Moreover, the use of so many tests through the manuscript (without pre-specified hypotheses) would require the use of some kind of adjustment.

Response: We agree with the reviewer that it is common practice to combine descriptive and inferential statistics (please refer to very many retrospective cohort studies, which are very similar compared to our investigation). We also did this here in a way to report and summarize the key characteristics of our cohort and subcohorts using descriptive statistics, while drawing first conclusions from our sample to a population using inferential statistics (95%CIs, correlations, etc.). For adjustment of age and other candidate covariates, respective multiple linear/logistic regression models were reported.

Moreover, we tried also to clearer formulate the obvious hypothesis from previous literature to be answered, namely that there are differences between SMA and ALS subcohorts regarding diaphragm functioning and thus ultrasound imaging results, as it was described for children with SMA.

Of note, however, most of the new inferential statistics during revision#1 were reported in direct response to the comments of the other reviewers, who are ultimately satisfied by our statistical reporting now. This was specifically concerning table 2 and 3. Thus we can’t remove all inferential statistics since 2 of 3 reviewers indeed wished to have us reporting these.

Nevertheless, we carefully went through the manuscript and through your comments from revision round 1 &2 once more and tried hard to simplify the results section wherever possible. Additionally, we now depict table three separatedly for the ALS and SMA subcohorts (as suggested in revision round #1). However, we still included heatmaps as partial correlations with adjustment for age as suggested by reviewers 1&2. The respective limitations are discussed in even more detail in the discussion section. We also tried hard to tone down the conclusions drawn. We would be very thankful if the reviewer is able to accept our approach as a very common approach in such investigations to be able to address the comments of the other reviewers sufficiently and to make it convenient for the potential readership of our manuscript to follow our results and discussion.

  • Since this is a retrospective study and not all patients performed all measurements, it should be clearly stated how many patients performed each measurement and when (was this on the same day of the US?). Otherwise, some results (such as correlations) cannot be adequately interpreted.

Response: We agree with the reviewer but like to draw his / her attention to the Supplemental Materials: The respective amount of patients, who underwent each single measurements were listed below every single correlation in the supplemental material already in revision#1 and are now as well depicted in the supplemental tables 1-3.

  • The main conclusion "we provide evidence for good applicability of diaphragm ultrasound imaging in routine clinical use for the determination / assessment of respiratory dysfunction of MND patients" is not supported by the data. Authors provide evidence for the viability of US imaging, but not for its "routine clinical use for the assessment of respiratory dysfunction". This is a mainly descriptive retrospective work that allows making some interesting hypotheses that should be confirmed in future prospective studies. I would overall recommend the authors to tone down their enthusiasm and to deepen the methodological limitations of the study, which are not sufficiently described.

Response: We deeply acknowledge the reviewer for his very positive and constructive comment. However, we do already stated that our results need further prospective approval. We already described a lot of methodological limitations (see Discussion section). Reviewers 1 and 2 are accordingly satisfied with our explanations. Nevertheless, we tried hard to tone down our enthusiasm (see colour marked text in re-revised manuscript).

  • I am aware that the authors may not agree with some of the points exposed above and I sincerely appreciate their effort to improve their manuscript. However, I consider that there are still serious methodological flaws.

Response: We are very grateful for the constructive discussion of the reviewer. We once more addressed every single concern raised by the reviewer. Of note, we want to point out that the other two reviewers already accepted the manuscript in its current form and that especially the inferential statistics and age adjustment for table 2&3 were recommended by them, who are now satisfied. Thus, the reviewer is solely raising „serious methodological flaws“. As mentioned above, we believe that our study is far from having serious methodological flaws, but of course can always be improved.

With the great help of the comments above, we believe that we indeed improved the manuscript significantly once more. We tried hard to find a way to reconcile the sometimes conflicting recommendations of the reviewers and hope that we found middle ground on which all three reviewers might agree and finally accept the manuscript. Once more thank you very much for your elaborated feedback helping us to improve our manuscript.